# The Thioredoxin System in *Edwardsiella piscicida* Contributes to Oxidative Stress Tolerance, Motility, and Virulence

**DOI:** 10.3390/microorganisms11040827

**Published:** 2023-03-24

**Authors:** Jiaojiao He, Su Liu, Qingjian Fang, Hanjie Gu, Yonghua Hu

**Affiliations:** 1School of Life Sciences, Hainan University, Haikou 570228, China; 2Institute of Tropical Bioscience and Biotechnology, Chinese Academy of Tropical Agricultural Sciences, Haikou 571101, China; 3School of Marine Sciences, Ningbo University, Ningbo 315211, China; 4Hainan Provincial Key Laboratory for Functional Components Research and Utilization of Marine Bio-Resources, Haikou 571101, China; 5Laboratory for Marine Biology and Biotechnology, Pilot National Laboratory for Marine Science and Technology, Qingdao 266071, China

**Keywords:** *Edwardsiella piscicida*, thioredoxin system, oxidative stress, motility, virulence

## Abstract

*Edwardsiella piscicida* is an important fish pathogen that causes substantial economic losses. In order to understand its pathogenic mechanism, additional new virulence factors need to be identified. The bacterial thioredoxin system is a major disulfide reductase system, but its function is largely unknown in *E. piscicida*. In this study, we investigated the roles of the thioredoxin system in *E. piscicida* (named TrxB_Ep_, TrxA_Ep_, and TrxC_Ep_, respectively) by constructing a correspondingly markerless in-frame mutant strain: Δ*trxB*, Δt*rxA*, and Δ*trxC*, respectively. We found that (i) TrxB_Ep_ is confirmed as an intracellular protein, which is different from the prediction made by the Protter illustration; (ii) compared to the wild-type strain, Δ*trxB* exhibits resistance against H_2_O_2_ stress but high sensitivity to thiol-specific diamide stress, while Δ*trxA* and Δ*trxC* are moderately sensitive to both H_2_O_2_ and diamide conditions; (iii) the deletions of *trxB_Ep_*, *trxA_Ep_*, and *trxC_Ep_* damage *E. piscicida*’s flagella formation and motility, and *trxB_Ep_* plays a decisive role; (iv) deletions of *trxB_Ep_*, *trxA_Ep_*, and *trxC_Ep_* substantially abate bacterial resistance against host serum, especially *trxB_Ep_* deletion; (v) *trxA_Ep_* and *trxC_Ep_*, but not *trxB_Ep_*, are involved in bacterial survival and replication in phagocytes; (vi) the thioredoxin system participates in bacterial dissemination in host immune tissues. These findings indicate that the thioredoxin system of *E. piscicida* plays an important role in stress resistance and virulence, which provides insight into the pathogenic mechanism of *E. piscicida*.

## 1. Introduction

The development of aquaculture plays a vital role in the rational utilization of land resources. Aquaculture accounts for approximately half of all fish production worldwide and remains one of the fastest-growing agricultural industries [1]. However, the emergence of edwardsiellosis has become one of the factors that jeopardize aquaculture, so there is an urgent need to combat this disease. *Edwardsiella* was initially described in 1962 and was established in 1965 by Ewing and his collaborators as a member of the *Enterobacteriaceae* family [2,3]. Currently, the *Edwardsiella* genus contains five species, including *E. tarda*, *E. ictaluri*, *E. hoshinae*, *E. anguillarum*, and *E. piscicida* [4]. *E. piscicida*, formerly known as *E. tarda*, is a Gram-negative, facultatively anaerobic, capsule-free, rod-shaped, motile, and intracellular pathogen that causes serious infections both in marine and freshwater fish, causing severe economic losses in the aquaculture industry [5,6]. In addition, *E. piscicida* has a wide host range, such as shellfish, amphibians, and reptiles. Diseases caused by *E. piscicida* infection have been reported in a lot of countries and regions [7,8,9].

Like other important pathogens, *E. piscicida* expresses many important virulence factors, which can help it escape the immune defenses of its host and achieve survival, reproduction, and infection in said host. The main virulence factors include a type III secretion system and a type VI secretion system, several two-component systems, an iron-absorption regulatory factor, hemolysin, a quorum-sensing system, and some environmental stress factors [4,7,10,11,12,13,14,15]. For example, *E. piscicida* adapts well to different pH, salinity, and temperatures [16,17,18]. All of these characteristics have made this bacterium a perfect model for elucidating the pathogenesis of intracellular pathogens [4]. Despite all of this, the pathogenesis of *E. piscicida* is still not very clear, and it is important and urgent to identify and analyze new unknown virulence factors.

The thioredoxin system is one of the two main antioxidant enzyme systems found in various organisms [19] and is ubiquitous in all living organisms [20]. The thioredoxin system is composed of thioredoxin (Trx), thioredoxin reductase (TrxR/TrxB), and reduced substrate nicotinamide adenine (NADPH). Trx, containing thioredoxin-A (TrxA) and thioredoxin-C (TrxC), is usually a 12 kDa thiol-disulfide oxidoreductase with a variety of functions, including reducing protein disulfide, maintaining proper functions of key regulators in motility, boosting the capability of pathogen to adhere to the epithelial cell, and affecting various virulence factors [21]. TrxR/TrxB is a dimeric enzyme belonging to the flavoprotein family of pyridine nucleotide–disulfide oxidoreductases. Members of this family are homodimeric proteins, each of which includes an FAD prosthetic group, an NADPH binding site, and an active site containing a redox-active disulfide [22]. Two forms of TrxR/TrxB have evolved, one in bacteria, archaea, and lower eukaryotes and another in higher eukaryotes [22]. In humans, TrxR is widely used as a target for cancer therapy, due to its role in regulating cellular redox balance as well as redox-mediated signal transduction [23], while in bacteria, TrxR/TrxB plays an important role by removing reactive oxygen species (ROS) in cells, thereby contributing to the resistance of bacteria to oxidative stress [24]; NADPH is responsible for transferring electrons to TrxR/TrxB to keep it in a reduced state [25]. The thioredoxin system is involved in enzyme function, DNA synthesis, gene transcription, cell growth and apoptosis, and the defense of oxidative stress [26]. However, the function of the thioredoxin system proteins is largely unknown in *E. piscicida*.

In this study, the thioredoxin system, TrxB_EP_, TrxA_Ep_, and TrxC_Ep_, of *E. piscicida* were identified, and their roles in adversity adaptation, motility, and pathogenicity were dissected. Our results present the first insights into the biological function of the thioredoxin system proteins in *E. piscicida*, which will contribute to comprehending *E. piscicida* pathogenesis.

## 2. Materials and Methods

### 2.1. Bioinformatics Analysis

Gene and protein information was obtained using NCBI (https://www.ncbi.nlm.nih.gov/ (accessed on 14 November 2022)). Jalview was used to analyze homologous sequence alignment. Protter (http://wlab.ethz.ch/protter/# (accessed on 14 November 2022)) was used to predict protein sequence features.

### 2.2. Bacteria, Plasmids and Cells

The bacteria and plasmids used in this study were described in Appendix A. *E. piscicida* TX01 was isolated from diseased Japanese flounder at fish farms in northern China and was determined by 16S rRNA gene analysis [27]. *E. coli* D1000 S17-λpir was purchased from Biomedal (Sevilla, Spain). *E. coli* DH5α was purchased from TransGen Biotech (Beijing, China). Bacteria were grown in Luria–Bertani medium (LB) broth or on LB agar at 37 °C (for *E. coli*) or 28 °C (for *E. piscicida*). When antibiotics were needed, the final concentrations of tetracycline (TC), chloramphenicol (CAP), polymyxin B (PMB), ampicillin (AMP), and kanamycin (KAN) in the medium were 15, 30, 100, 100, and 50 μg/mL, respectively. The RAW264.7 cells were propagated at 37 °C in a 5% CO_2_ incubator in DMEM supplemented with 10% fetal bovine serum.

### 2.3. Animals and Ethics

Clinically healthy tilapias were purchased from local commercial fish farms and raised in aerated fresh water at about 26 °C as previously reported [28]. Clinically healthy female mice (4–6-week-old) were purchased from a local commercial mouse breeding base and housed at about 26 °C. No mouse died before meeting the criteria for humane endpoint euthanization during this study. The study was approved by the ethics committee of the Institute of Tropical Bioscience and Biotechnology, Chinese Academy of Tropical Agricultural Sciences. Efforts were taken to ensure that all research animals received good care and humane treatment. Animals were euthanized followed by cervical dislocation when the experimental endpoints were reached.

### 2.4. Construction of Mutant Strain and Complementary Strain

The in-frame deletion mutant strains used in this study were constructed as described previously [29]. The primers used in this study were shown in Appendix A. The in-frame deletion mutant was constructed by double-crossover allelic exchange using suicide vector pDM4. To construct the mutant strain Δ*trxA*, primers TrxAKOF1/TrxAKOR1 and TrxAKOF2/TrxAKOR2 were used for the upstream and downstream PCR of *trxA*. The fusion PCR products amplified by primer pair TrxAKOF1/R2 were cloned into the pDM4 at the *Bgl* II site, in which a 1330 bp segment in-frame deletion was created, resulting in pDMTrxA. S17-1 λpir was transformed with pDMTrxA, and the transformants were conjugated with TX01. The transconjugants were selected on TSA plates supplemented with 10% sucrose. The mutations of Δ*trxB* and Δ*trxC* were made with the same operation application.

### 2.5. Expression of Recombinant TrxB and Determination of Enzyme Activity

In order to express the recombinant TrxB (rTrxB) protein, segments of *trxB* were amplified with the primer pair TrxBF/R, and then the PCR segments were cloned into pET28a with double-enzyme digestion (*Bam*HI + *Hind*III), resulting in pET-TrxB. The recombinant plasmid pET-TrxB was extracted and transformed into *E. coli* BL21(DE3). rTrxB_Ep_ was expressed and purified as described previously [30]. In brief, *E. coli* BL21(DE3) carrying pET-TrxB was cultured to an OD_600_ of 0.5 in LB broth in the presence of 25 µg/mL kanamycin, and then 0.1 mM IPTG was added to the broth. After 6 h of induction, the preparation of the lysate and the purification and refolding of the recombinant protein were conducted as follows: the *E. coli* BL21(DE3) carrying pET-TrxB were incubated with Lysis Buffer B (100 mM NaH_2_PO_4_, 10 mM Tris-Cl, and 8 M urea [pH 8.0]) at 20 °C for 2 h and the protein extract was obtained after centrifugation. The recombinant protein was purified by Ni-NTA agarose (QIAGEN, Germany) under denaturing conditions. The purified recombinant protein was dialyzed in a buffer containing 50 mM Tris-Cl (pH = 8.5), 50 mM NaCl, 2 mM reduced glutathione, 5 mM EDTA, 5 mM β-mercaptoethanol, 20% glycerol, and gradually decreasing urea. At the same time, a control protein, rOmpR_Ep_ (unpublished data), was obtained through the same experimental process.

The content of rTrxB_Ep_ and rOmpR_Ep_ (control) was measured using the BCA Protein Assay Kit (Solarbio, Beijing, China). The thioredoxin reductase activity of the rOmpR_Ep_ and rTrxB_Ep_ was determined by using Micro Oxidized Thioredoxin Reductase Assay Kit (Solarbio, Beijing, China) according to the manufacturer’s instructions and the description by Zheng et al. [31]. Thioredoxin reductase catalyzes the reduction of DTNB by NADPH to produce TNB and NADP^+^. TNB has a characteristic absorption peak at 412 nm, but reduced glutathione can also react with DTNB to produce TNB. Therefore, this kit uses 2-vinyl pyridine to inhibit reduced glutathione in the sample and calculates the activity of thioredoxin reductase by measuring the increase rate of TNB at 412 nm expressed as U/mg protein. The experiment was repeated three times.

### 2.6. Preparation of Antibody and ELISA

The antibody of TrxB_Ep_ was prepared as described previously [30]. Freund’s complete and incomplete adjuvant were purchased from Sigma-Aldrich. The detection of antibody titer was determined by indirect ELISA [32].

### 2.7. Preparation of Cellular Component Protein and Western Blotting

*E. piscicida* was inoculated into 50 mL DMEM containing polymyxin B at 1% inoculation and was cultured at 28 °C until OD_600_ = 0.5. Then, the bacteria were incubated at 0 °C for 10 min, and the final concentration of 1 mM phenylmethylsulfonyl fluoride (PMSF) was added. After mixing, the mixture was centrifuged by 5000× *g* for 10 min at 4 °C, and the supernatant and precipitation were collected, respectively. The supernatant was filtered by 0.22 μm microporous membrane and added into a 10 kDa ultrafiltration tube. The supernatant was concentrated to 200 μL by centrifugation, and finally, the extracellular protein was obtained. The extraction of whole-cell protein, periplasmic protein, and intracellular protein was conducted as described in a previous report [33]. The obtained proteins were subjected to SDS-PAGE. The antibody of TrxB_Ep_ and Goat anti-Mouse IgG HRP secondary antibody (TransGen Biotech, Beijing, China) were used for Western blotting as described previously [30].

### 2.8. Resistance to Environmental Stress

The bacterial growth curve was carried out in 96-well plates. *E. piscicida* strains were cultured in LB broth medium until the exponential phase, then bacteria were gradient diluted to 10^5^ CFU/mL in fresh LB medium supplemented with 300 μM H_2_O_2_, or with 550 μM diamide, or with pH = 5, or with 100 μM 2,2’-Bipyridine and were cultured at 28 °C with gentle shaking. Using the Bioscreen C Automated Growth Curve System (OyGrowth Curves Ab Ltd., Finland), the absorbance at 600 nm was measured and the growth curve was monitored at 2 h intervals. Under the challenge of 300 μM H_2_O_2_ or 550 μM Diamide, *E. piscicida* strains were diluted in different concentrations, dropped on LB agar plates, and incubated at 28 °C for 48 h. For the survival experiment, bacteria in the logarithmic growth phase were diluted and treated with 300 μM H_2_O_2_ or 550 μM diamide for 1 h, and then the colony-forming units were counted. The survival rate was calculated as follows: (number of each strain with pressure treatment)/(number of each strain without pressure treatment) × 100%. The experiment was repeated three times.

### 2.9. Measurements of Intracellular Reductive Capacity in Cell Extracts

The thioredoxin reductase activity assay was referred to Ana Paunkov [34]. In short, *E. piscicida* strains were cultured in LB medium, supplemented with 30 μM H_2_O_2_ or 50 μM diamide overnight at 28 °C, and 3 mL of 5 h cultures were centrifuged in order to obtain the pellets. Pellets were resuspended in a 1 × PBS buffer containing 0.5% Triton X-100 and were lysed by an ultrasonic disrupter (SCIENTZ, China). Insoluble material was removed by centrifugation for 10 min at 12,000× *g* at 4 °C. The supernatants were collected and the protein concentrations were determined. The intracellular reductive capacity of cell extract was measured at OD_412_ by determining the reduction of DNTB in reaction buffer containing 100 mM KH_2_PO_4_, 0.2 mM NADPH, 1 mM DTNB, and 100 μL of cell extract. The experiment was repeated three times. The reductive capacity was calculated as follows: (data of each strain with pressure treatment)/(data of each strain without pressure treatment) × 100%. 

### 2.10. Transmission Electron Microscopy

Bacteria grown to an OD_600_ of 0.5 were collected by centrifugation (3500× *g* for 10 min) and softly resuspended in PBS. Suspensions of different strains were dropped onto the copper grid and were allowed to stand for 20 min with the purpose of forming thin films on the copper grids. Subsequently, the excess solutions were removed, and the copper grids were dried naturally. Ultimately, all samples were tested by HT7700 biological TEM (Hitachi, Japan) [35].

### 2.11. Motility Assay

The motility assay was performed as described previously [36]. Briefly, 2 μL of bacteria suspension were spotted onto the center of fresh swimming plates (0.3% agar) or swarming plates (0.6% agar). Then, the plates were incubated at room temperature and bacterial motility was observed. The experiment was performed three times.

### 2.12. Bacterial Resistance to Non-Immune Fish Serum

The experiment on the bacterial resistance to non-immune fish serum was performed as described previously [37]. Briefly, bacteria in the exponential phase were washed three times with PBS. Next, 10 μL of bacteria (including approximately 1 × 10^5^ CFU) were incubated with 50 μL non-immune fish serum or PBS (control) for 60 min. The mixtures were serially diluted and plated in triplicate on LB agar plates. The plates were incubated at 28 °C for 24 h, then the number of colonies was determined. The survival rate was calculated as follows: [(number of serum-treated cells)/(number of control cells)] × 100%. The experimental procedures were performed three times.

### 2.13. Bacterial Invasion of Host Cells

The bacterial replication in RAW264.7 cells was carried out as previously described [37]. RAW264.7 (1 × 10^5^) cells were cultured in 96-well plates in a 5% CO_2_ incubator at 28 °C, and different *E. piscicida* strains, WT, Δ*trxA*, Δ*trxB*, and Δ*trxC*, (1 × 10^6^ CFU) were added to RAW264.7 cells, which were cultured in DMEM medium (Gibco, Grand Island, NY, USA) containing 10% FBS (Gibco, Grand Island, NY, USA). After incubation at 28 °C for 2 h, cells were gently washed with PBS and incubated with fresh DMEM containing 200 μg/mL gentamicin (the minimum inhibition concentration of gentamicin is 20 μg/mL) for 2 h in order to kill extracellular bacteria. After washing with PBS, the cells were cultured in fresh DMEM containing 10 μg/mL gentamicin for different time points. In order to detect the number of viable bacteria in RAW264.7 cells at different time points, as mentioned above, the cells were lysed and evenly coated on LB agar plates, and then cultured at 28 °C for 24 h. The experiment was repeated three times.

### 2.14. Bacterial Dissemination in Fish Tissues

Healthy tilapias (average weight 14.5 g) were purchased from a commercial fish farm in Haikou and fed in aerated water at 26 °C for 2 weeks. Before the experiment, fish were randomly sampled in order to examine whether there were bacteria in the blood, liver, spleen, and kidney. For tissue dissemination analysis, different *E. piscicida* strains were cultured in LB broth medium to an OD_600_ of 0.5, the cells were washed with PBS and resuspended in PBS to 10^7^ CFU/mL. Tilapias were randomly divided into 5 groups with 10 fish per group, infected with 50 μL of each bacterial suspension or PBS (control) by intramuscular injection. At 24 h and 48 h post-infection, excessive MS-222 (Sigma-Aldrich, St. Louis, MO, USA) was used for euthanasia on fish. The spleen and kidney tissues were taken under sterile conditions and bacteria recoveries was analyzed by plate counting [38]. The experiment was repeated three times.

### 2.15. Statistical Analysis

Analysis of variance (ANOVA) was performed on all data using SPSS 23 software (SPSS Inc., Chicago, IL, USA). For multigroup comparisons with Gaussian distribution, one-way ANOVA with Tukey–Kramer’s multiple-comparison test was used after the confirmation of homogeneity of variance among the groups by Bartlett’s test. For multigroup comparisons with non-Gaussian distribution, a Kruskal–Wallis test with Dunn’s test was used. Data are presented as the means ± SEMs (N = 3). N, the number of times the experiment was performed. *p* values were obtained by analysis of variance using SPSS 23.

## 3. Results

### 3.1. Bioiformatics Analysis of Trx System in E. piscicida

*E. piscicida* TrxB, TrxB_Ep_ (ETAE_2202), is a 323-amino-acid protein encoded by a 972 bp open reading frame (ORF). Bioinformatics analysis reveals that the predicted molecular weight of TrxB_Ep_ is 34.59 kDa, and the isoelectric point of TrxB_Ep_ is 5.47. *E. piscicida* TrxA, TrxA_Ep_ (ETAE_0099) contains a 327 bp ORF, which codes 108 amino acid residues with a calculated molecular mass of 11.59 kDa and a theoretical pI of 5.04. *E. piscicida* TrxC, TrxC_Ep_ (ETAE_0559) has a 143 bp ORF, which codes 143 amino acid residues with a calculated molecular mass of 15.80 kDa and a theoretical pI of 5.31. As shown in Figure 1, TrxA_Ep_, TrxB_Ep_, and TrxC_Ep_ share high amino acid sequence identities with homologues of many Gram-negative bacteria, including *H. alvei*, *E. coli*, *S. enterica*, and *L. richardisequence*; however, they share low sequence identities (36.59%, 40.74%, and 19.58%, respectively) with the Gram-positive stain *L. monocytogenes*.

### 3.2. The Location of Trx System and the Enzyme Activity of TrxB_EP_

The thioredoxin system plays a central role in intracellular redox maintenance in most cells [39]. However, the Protter illustration shows that TrxB_Ep_ has a signal peptide and is predicted to be extracellularly located, although TrxA_Ep_ and TrxC_Ep_ are predicted to be intracellular (Figure 2A). In order to observe the distribution of TrxB_Ep_, whole-cell protein, secreted protein, periplasmic protein, and the intracellular protein of *E. piscicida* were prepared (Figure 2B); meanwhile, rTrxB_Ep_ was expressed and purified (Appendix A) and antiserum against rTrxB_Ep_ was prepared (Appendix A). Western blot analysis displayed that the target band was observed in the whole-cell protein and intracellular protein but not in the secreted protein and periplasmic protein (Figure 2C). These results suggest that TrxB_Ep_ is an intracellular protein.

Next, the thioredoxin reductase activity of rTrxB_Ep_ was determined using the Micro Oxidized Thioredoxin Reductase Assay Kit. The analysis of thioredoxin reductase activity based on OD_412_ showed that the absorbance induced by 20 μL of rOmpR_Ep_ and rTrxB_Ep_ was 0.0000 and 0.0066, which corresponded to the thioredoxin reductase activity of 0.00 and 2.54 U/mg protein, indicating that TrxB_Ep_ is a thioredoxin reductase (Appendix A).

### 3.3. Construction of Mutant Strains of Trx System

In order to further study the function of the thioredoxin system, we constructed three mutants—Δ*trxA*, Δ*trxB*, and Δ*trxC*—by deleting the gene segment from 64 to 234 of *trxA_Ep_*, 115 to 858 of *trxB_Ep_*, and 103 to 319 bp of *trxC_Ep_* (Appendix A).

### 3.4. The Trx System Is Involved in the Resistance against Oxidative Stress, Acid Stress, and Iron Deficiency Stress

Under normal conditions, Δ*trxB* grows a little slower than WT, and the growths of Δ*trxA* and Δ*trxC* have no significant difference with WT (Figure 3[A1])**.** The intracellular reductive capacities of WT and three mutants have no significant differences when cultured in normal LB broth medium (Figure 3[A2]). These results suggest that the deletion of *trxA_Ep_*, *trxB_Ep_*, or *trxC_Ep_* basically does not affect the growth and reductive capacity of *E. piscicida* under normal conditions. As the Trx system plays a central role in maintaining intracellular redox, we compared the roles of the Trx system in resistance H_2_O_2_ and diamide. When grown in solid LB medium containing H_2_O_2_, Δ*trxB* appeared to have similar growth, but Δ*trxA* and Δ*trxC* presented sluggish growth compared to WT (Figure 3[B1]). However, when grown in LB liquid medium containing H_2_O_2_, the growth of the four strains is similar (Figure 3[B3]). When temporarily exposed to a high concentration of H_2_O_2_, the intracellular reductive capacities of the three mutants were basically equal but were significantly lower than the WT (Figure 3[B2]); the survival rate of Δ*trxA* (55%) was comparative to Δ*trxC* (62%), and both were significantly lower than the WT (73%), which is also significantly lower than Δ*trxB* (109%) (Figure 3[B4]). When the oxidant was diamide, Δ*trxB* hardly grew in LB broth medium or LB agar medium, while Δ*trxC* was basically similar to WT, and Δ*trxA* was slightly slower than WT (Figure 3[C1,C3]). When temporarily exposed to a high concentration of diamide, the intracellular reductive capacities of all three mutants were lower than the WT (Figure 3[C2]), the survival rates of Δ*trxB* (58%) were comparative to Δ*trxC* (55%), and both were significantly lower than the WT (72%), though that of Δ*trxA* (14%) was dramatically lower than those of Δ*trxB* and Δ*trxC* (Figure 3[C4]). These results indicate that TrxB_Ep_, TrxA_Ep_, and TrxC_Ep_ play an important role in bacterial resistance against different types of oxidation stress.

In addition to oxidation pressure, acid stress and iron deficiency stress were also used in the resistance experiment, and the results indicated that Δ*trxB* and Δt*rxA* exhibit weaker growth than WT under acid conditions (pH = 5) and iron deficiency conditions (Figure 3D), while Δ*trxC* is basically similar to WT. The results collectively suggest that the Trx system of *E. piscicida* participates in various adverse circumstances, especially oxidative stress.

### 3.5. The Trx System Is Essential for Bacterial Motility and Flagellum Formation

In this study, we explored the involvement of the thioredoxin system in bacterial mobility by detected bacterial swimming, and the result showed that the motility zone diameters of Δ*trxB* (average diameter 19 ± 3 mm), Δ*trxA* (average diameter 16 ± 1 mm), and Δ*trxC* (average diameter 22 ± 1 mm) were significantly smaller than that of the WT (average diameter 27 ± 1 mm) (Figure 4A). At the same time, bacterial swarming was examined, and the result showed that the branching motions of Δ*trxA* and Δ*trxB* were obviously weaker than that of the WT, but Δ*trxB* completely lost such motion (Figure 4B). In order to explore whether decreased motility was correlated with flagellum formation, morphological observations were performed, and the results showed that there were a great quantity of flagella around WT and Δ*trxC*, a few flagella around Δ*trxA*, and no flagella around Δ*trxB* (Figure 4C). These results indicate that the Trx system participates in *E. piscicida*’s flagella formation and motility, and *trxB_Ep_* plays a decisive role.

### 3.6. The Trx System Is Involved in Bacterial Resistance against Non-Immune Fish Serum and Bacterial Survival in Host Phagocytes

Resistance against host serum killing is an important virulence characteristic of *E. piscicida* [40], thus we examined the effect of the thioredoxin system. The results show that three Trx mutants exhibited remarkably lower survival rates than the WT, with the lowest-ranking strain being Δ*trxB* (27%), which was much lower than Δ*trxA* and Δ*trxC* (about 50%) (Figure 5A). This result suggests that the thioredoxin system is involved in bacterial resistance against serum killing.

Next, we investigated the role of Trx system in the survival and replication of *E. piscicida* in host phagocytes. The result showed that the amounts of WT and Δ*trxB* from RAW264.7 cells was basically the same at the three examined time points, but the amounts of Δ*trxA* were significantly less than WT at these three examined time points, while Δ*trxC* was less than WT only at 4 h post-infection (Figure 5B). This result indicates that the thioredoxin system of *E. piscicida*, except *trxB_Ep_*, is involved in bacterial survival and replication in host phagocytes.

### 3.7. The Trx System Participates in Bacterial Dissemination in Host Tissues

In order to explore the involvement of the thioredoxin system in bacterial virulence in vivo, tilapia were infected by different strains, then the invasion of immune tissues by bacteria was determined. The results showed that the counts of the bacteria from the spleen of three mutant-strain-infected tilapia were significantly lower than those of WT-infected tilapia at two examined time points, and a similar result was observed from the kidney of treated tilapia (Figure 6). Furthermore, we did not observe fish mortality during the experiment because of the short duration. According to our laboratory data, infections with the same amount of WT generally show mortality 4–5 days after injection [28,37]. These results suggest that the Trx system of *E. piscicida* is correlated with bacterial dissemination in host tissues.

## 4. Discussion

Oxidative stress is one of the most susceptible pressures during bacterial growth and reproduction [41]. The latest research reveals that ROS in *E. tarda* are related to bacterial resistance to antibiotics and antibiotic-mediated killing efficacy [42]. The thioredoxin system is one of the major disulfide reductase systems used by bacteria against oxidative stress [43,44]. The function of the thioredoxin system in *E. piscicida* remains mostly unknown so far. In this study, several proteins (TrxA_Ep_, TrxB_Ep_, and TrxC_Ep_) included in the thioredoxin system of *E. piscicida* were identified, and their functions were investigated. Sequence analysis indicated that TrxA_Ep_, TrxB_Ep_, and TrxC_Ep_ all share a substantial portion of their identities with many homologous Gram-negative bacteria. It should be emphasized that those three proteins possess the CXXC motif, which is a characteristic active site employed by many redox proteins for the formation, isomerization, and reduction of disulfide bonds and for other redox functions [45]. TrxB_Ep_ is predicted to be extracellular, but our results confirm that it is an intracellular protein, which is consistent with the results of other microorganisms [46].

The thioredoxin system plays a crucial role in maintaining internal redox homeostasis [26]. For example, in *E. coli*, the *trxB*, *trxA*, and *trxC* mutants exhibit increased sensitivity compared with the wild-type strain when exposed to direct oxidant hydrogen peroxide in the stationary phase [47,48]. In *L. monocytogenes*, although hydrogen peroxide does not affect bacterial growth when *trxA* is deleted, *trxB* is significantly induced when bacteria are treated with H_2_O_2_ [49]. Consistently, our results showed that the deletion of *trxA* or *trxC* negatively affects the growth and survival rate of *E. piscicida* when in the presence of hydrogen peroxide. However, *trxB* mutation exhibits a contrary result, namely better growth and higher survival rate than the wild strain under H_2_O_2_ stress. Similar results have been reported in *Neisseria gonorrhoeae* and *E. coli*, in which the *trxB* mutant strains exhibit significantly higher resistance to H_2_O_2_ stress [47,48]. While the intracellular reductive capacity of the t*rxB* mutant is similar to Δ*trxA* and Δ*trxC*, which is to say that it is less than the wild strain, we speculate that the deficiency of thioredoxin reductase TrxB probably activates an unknown catalase. Such an interesting result inspires us to explore another oxidant, diamide, which is a thiol-specific oxidant. For example, in *Bacillus subtilis* and *Staphylococcus aureus*, diamide has been shown to induce the expression of *trxB* [40,50]. In *E. coli*, *S. aureus*, and *L. monocytogenes*, the expressions of *trxA* and *trxC* also rise when challenged with diamide [39,49,51]. Diamide is shown to oxidize glutathione (GSH) to the disulfide (GSSG) [52]. As expected, we found that under diamide stress, the deletion of *trxB_Ep_* tremendously reduced bacterial growth, whether in solid or liquid medium. In the presence of diamide, the intracellular reductive capacities of three mutants were consistent and were all lower than the wild strain, while the survival rate of three mutants were inconsistent—Δ*trxA* was remarkably lower than Δ*trxB* and Δ*trxC*, despite the latter two being lower than WT. The difference between the growth curve and survival rate may be caused by the discrepancy between continuous and transient oxidation stress. These findings illustrate that the thioredoxin system of *E. piscicida* is an important factor in bacterial stress tolerance, especially to oxidative stress. However, the mechanism through which the thioredoxin system responds to different types of oxidants should be explored in the future.

Studies of the effect of the thioredoxin system on motility have mainly focused on *trxB* [25,53]. In this study, three components of the Trx system were investigated, and the results showed that the deletion of *trxB_Ep_*, *trxA_Ep_*, and *trxC_Ep_* all weaken the motility of *E. piscicida*. In particular, the swarming of Δ*trxB* is almost completely irradicated, and, consistently, Δ*trxB* has lost its flagella, which indicates that *trxB_Ep_* is indispensable for flagellar formation and motility in *E. piscicida*. Similarly, in *E. coli*, a motility assay has shown that the mutation of *trxB* caused a strong inhibition of swarming but not of swimming [53]. It is unlikely that, in *L. monocytogenes*, *trxA* is essential for bacterial motility by maintaining the reduced intracellular monomer status of MogR, the key regulator for flagellar formation [49]. These reports and our results indicate that the Trx system is requisite for bacterial motility.

Reports indicate that the thioredoxin system plays an important role in bacterial virulence [54,55,56]. For example, in the plant pathogen *Fusarium graminearum*, thioredoxin reductase is required for virulence [57]. In the human pathogen *Aspergillus fumigatus*, thioredoxin reductase is also important for its successful infection [58]. Likely, the lack of TrxA in *Acinetobacter baumannii* is associated with decreased expression of type IV pili-related genes and attenuated virulence [21]. Similarly, in our case, the mutation of *trxA_Ep_* or *trxC_Ep_* lead to reduced survival and replication in host phagocyte, despite the mutation of *trxB_Ep_* not having any effect. At the individual level, the deletions of three genes led to a significant decrease in the ability of *E. piscicida* to infect tilapia tissues. One of the important reasons for this is that the deficiency of the thioredoxin system led to a considerable reduction in resistance to bacterial serum and host oxidative killing. These results illustrate that the thioredoxin system plays an important role in the virulence of *E. piscicida*.

In conclusion, we investigated for the first time the function of the thioredoxin system in *E. piscicida*. The system is essential to resisting the oxidative stress induced by a direct oxidant such as hydrogen peroxide and the thiol-specific oxidizing agent diamide. The thioredoxin system also plays a role in bacterial motility and flagellum formation. More importantly, the thioredoxin system is required for the pathogenicity of *E. piscicida*. These findings support the conclusion that the thioredoxin system is a stress resistance factor and virulence factor of *E. piscicida*, which provides insights into the pathogenic mechanism of *E. piscicida*.

## Figures and Tables

**Figure 1 microorganisms-11-00827-f001:**
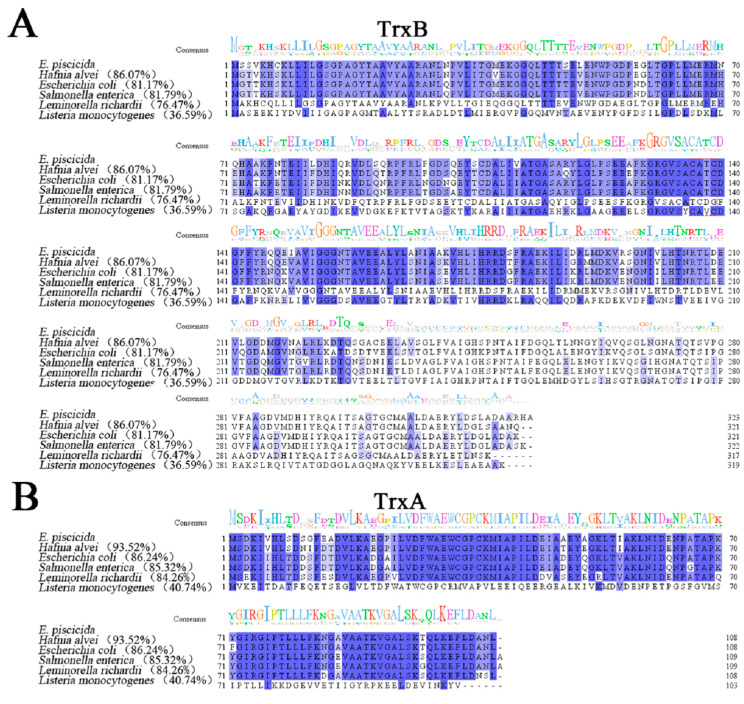
Multiple sequence alignments of TrxB (**A**), TrxA (**B**), and TrxC (**C**) with their homologues. The consensus amino acid residues are shown in blue, and the amino acid residues with conservative degree higher than 75% are shown in light blue. A consensus sequence logo has been created using Jalview. The GenBank accession numbers of the TrxB homologues are as follows: *Edwardsiella piscicida*, WP_012849033.1; *Hafnia alvei*, WP_149226215.1; *Escherichia coli*, NP_415408.1; *Salmonella enterica*, AGK67561.1; *Leminorella richardii*, SQI39560.1; *Listeria monocytogenes*, WP_003722610.1. The GenBank accession numbers of the TrxA homologues are as follows: *E. piscicida*, WP_012846980.1; *H. alvei*, WP_115349310.1; *Escherichia coli*, HAW3242924.1; *S. enterica*, AGK68906.1; *L. richardii*, WP_111741836.1; *L. monocytogenes*, NP_464758.1. The GenBank accession numbers of the TrxC homologues are as follows: *E. piscicida*, WP_012847430.1; *H. alvei*, WP_130998671.1; *E. coli*, NP_417077.1; *S. enterica*, NP_461584.1; *L. richardii*, WP_111739434.1; *L. monocytogenes*, WP_003723835.1.

**Figure 2 microorganisms-11-00827-f002:**
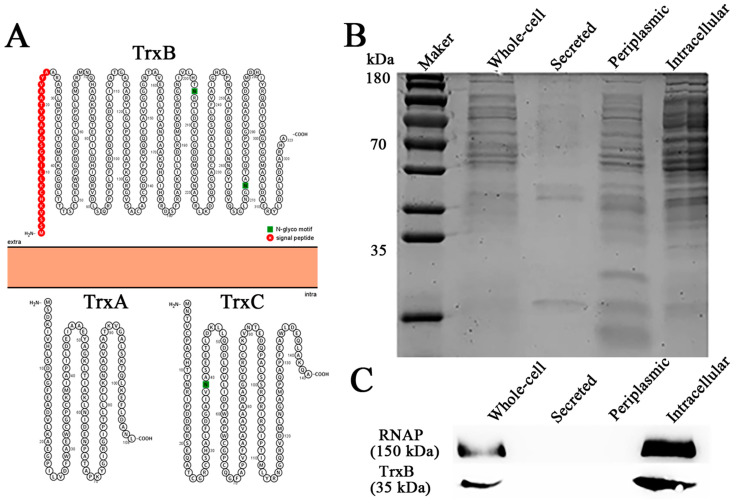
Cellular localization of TrxB_Ep_. (**A**) Protter illustration of TrxB_Ep_, TrxA_Ep_ and TrxC_Ep_. (**B**) protein components of *Edwardsiella piscicida* cells. (**C**) The distribution of TrxB_Ep_ tested by Western blot.

**Figure 3 microorganisms-11-00827-f003:**
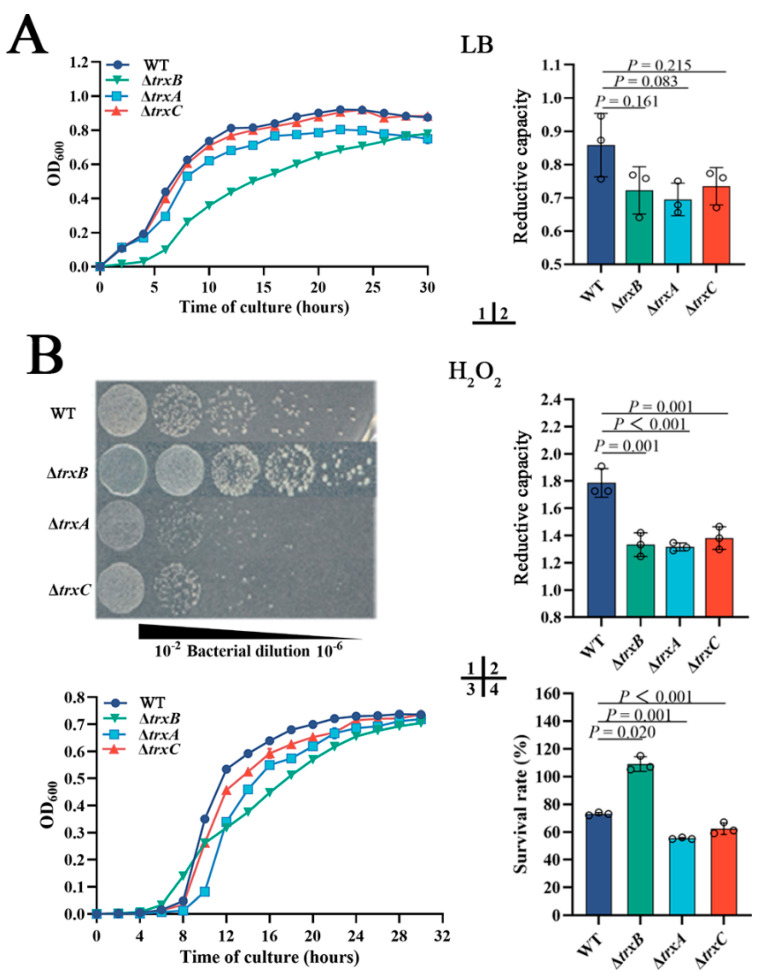
The roles of the Trx system in bacterial adversity resistance. (**A**) WT, Δ*trxB*, Δ*trxA*, and Δ*trxC* were cultured in LB broth, and then the cell density was measured at OD_600,_ and the intracellular reductive capacity was detected. (**B**) Bacterial growths on LB agar plates (1) and in liquid LB (3) with 300 μM H_2_O_2_; bacteria in the logarithmic growth phase were challenged with 30 mM H_2_O_2_ for 5 h, and the intracellular reductive capacity was determined (2). Bacteria in the logarithmic growth phase were diluted to 1:1000 and treated with PBS containing 300 mM H_2_O_2_ for 1 h, then the amount of viable bacteria was determined (4). (**C**) bacterial growths on LB agar plates (1) and in liquid LB (3) with 550 mM diamide; bacteria in the logarithmic growth phase were challenged with 50 mM diamide for 5 h, and intracellular reductive capacity was detected (2). Bacteria in the logarithmic growth phase were diluted to 1:1000 and treated with PBS containing 550 mM diamide for 1 h, then the content of viable bacteria was determined (4). (**D**) Strains were cultured in LB broth containing acid pressure (pH = 5) (1) or iron deficiency stress (100 μM Dp) (2) and incubated at 28 °C for 48 h. Data are expressed as means ± SEM (N = 3). N, the number of experiments performed. *p* values were obtained by analysis of variance using SPSS 23.

**Figure 4 microorganisms-11-00827-f004:**
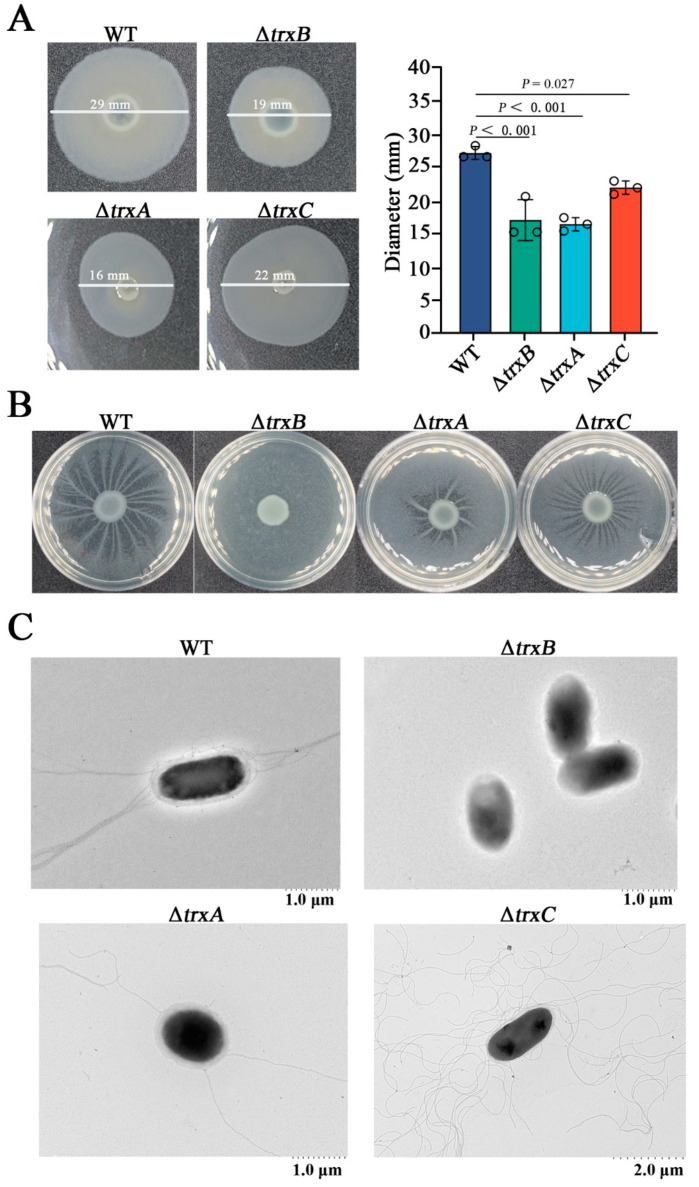
The effects of the thioredoxin system mutations on bacterial motility and flagellum formation. (**A**) The swimming of *Edwardsiella piscicida*. WT, Δ*trxB*, Δ*trxA*, and Δ*trxC* were cultured in LB medium to an OD_600_ of 0.6, and 1 μL cell suspensions were spotted onto the center of swimming plates containing LB medium plus 0.3% (*w/v*) agar. The plates were incubated at 28 °C for 24 h, and the motility zone diameter was measured. (**B**) The swarming of *E. piscicida*. Bacteria as described above were spotted onto the center of swimming plates containing LB medium plus 0.6% (*w/v*) agar and were incubated at 28 °C for 24 h. (**C**), the flagellum observation of *E. piscicida*. WT, Δ*trxB*, Δ*trxA* and Δ*trxC* were grown in LB medium, and the flagella were observed using the TEM. Data are presented as the means ± SEM (N = 3). N, the number of times the experiments were performed. *p* values were obtained by analysis of variance using SPSS 23.

**Figure 5 microorganisms-11-00827-f005:**
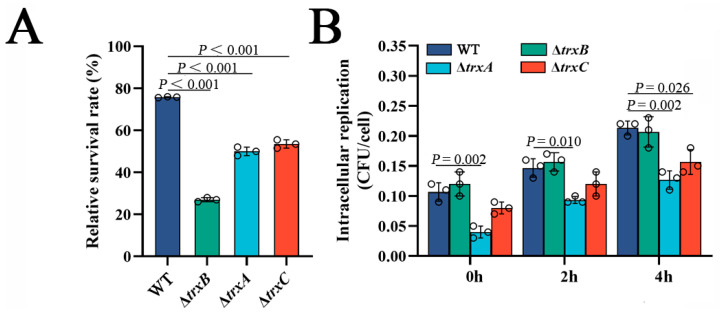
**The** effects of thioredoxin system mutations on bacterial survival in host serum and host macrophages. (**A**) The survival rate of WT, Δ*trxB*, Δ*trxA* and Δ*trxC* against non-immune fish serum. Strains in the early logarithmic phase were incubated with non-immune tilapia serum or PBS (control) for 1 h. The number of viable bacteria was determined. (**B**) *Edwardsiella piscicida* replication in macrophages. The murine macrophage cell line RAW264.7 was incubated with WT, Δ*trxB*, Δ*trxA* and Δ*trxC* for 2 h. After killing and washing extracellular bacteria, the macrophages were cultured for different lengths of time. At each time point, the viable intracellular bacteria were determined. Data are presented as the means ± SEM (N = 3). N, the number of times the experiments were performed. *p* values were obtained by analysis of variance using SPSS 23.

**Figure 6 microorganisms-11-00827-f006:**
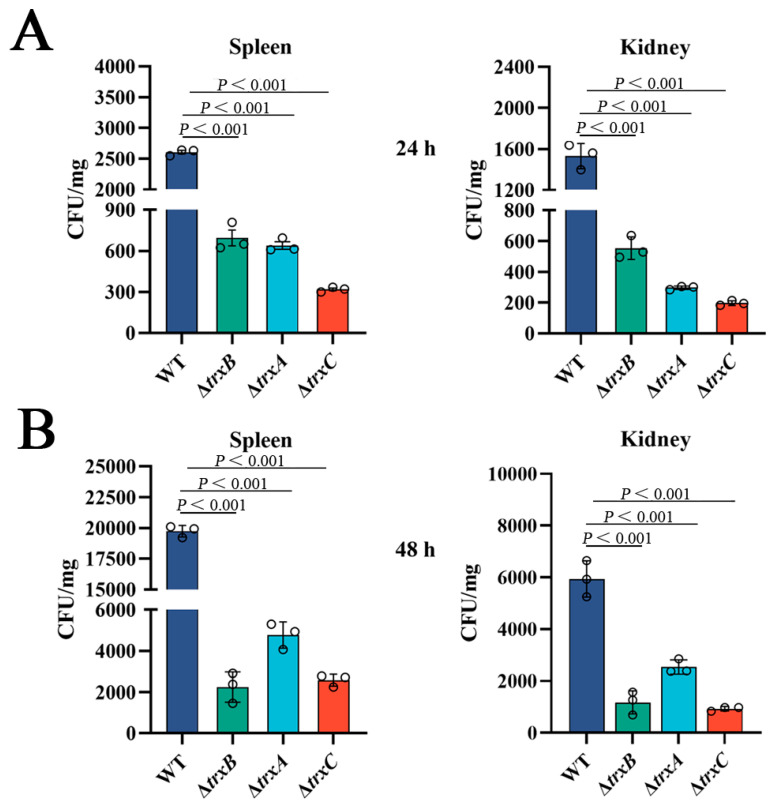
Effects of the thioredoxin system mutations on bacterial virulence. (**A**) *Edwardsiella piscicida* strains (WT, Δ*trxB*, Δ*trxA*, and Δ*trxC*) were used to infect tilapias, then, after 24 h, bacteria recovery from immune tissues (spleen and kidney) was analyzed by plate counting. (**B**) 48 h post-infection, bacteria were counted by plate counting. Data are presented as the means ± SEM (N = 3). N, the number of times the experiments were performed. *p* values were obtained by analysis of variance using SPSS 23.

## Data Availability

The data in this study are readily available upon reasonable request to the corresponding author.

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
