# Peer review of "The Thioredoxin System in Edwardsiella piscicida Contributes to Oxidative Stress Tolerance, Motility, and Virulence"

_microorganisms, 2023, doi:10.3390/microorganisms11040827_

Round 1
Reviewer 1 Report
The manuscript entitled “The thioredoxin system in Edwardsiella piscicida contributes to oxidative stress tolerance, motility, and virulence” is an important study as major virulence factors include the secretion system and iron absorption factor regulator, and it is important and urgent to identify and analyze new unknown virulence factors.
The introduction is relevant but must include new references.
The discussion, in the light of results and knowledge, is relevant.
Your manuscript will not be accepted unless both the technical and grammatical revisions have been made successfully.
Based on these comments, I recommend a moderate revision of analytical aspects of this manuscript before final decision about its acceptance.
Moderate comment:
Introduction:
Rewrite this part with new references it is a very old research (e.g., Leotta, G.A., 2009 and Jiao, X.D., et al., 2010)
See reference: Recent study (Parrino V. et al., 2022-2018) showed the role in order to assess the effects of environmental pollution.
It’s necessary to insert new references about the use of natural water see below reference:
Basova Marina, Svetlana Krasheninnikova, Parrino Vincenzo - Intra-Decadal (2012–2021) Dynamics of Spatial Ichthyoplankton Distribution in Sevastopol Bay (Black Sea) Affected by Hydrometeorological Factors. ANIMALS; 12, 3317; https://doi.org/10.3390/ani12233317; (2022).
Parrino Vincenzo, Cappello Tiziana, Costa Gregorio, Cannavà Carmela, Sanfilippo Marilena, Fazio Francesco, Fasulo Salvatore - Comparative study of haematology of two teleost fish (Mugil cephalus and Carassius auratus) from different environments and feeding habits. - THE EUROPEAN ZOOLOGICAL JOURNAL, vol. 85:1, p. 194-200, ISSN: 2475-0263, doi: 10.1080/24750263.2018.1460694; (2018).
2. Materials and methods:
2.8. Resistance to environmental stress
L167-227… For the survival experiment, bacteria in the logarithmic growth phase were diluted and treated with 300 μM H2O2 or 550 μM diamide for 1 h, and then the colony-forming unites were counted. the data was collected (the experiment protocol was developed by you with due care ? There was it mortality?
Statistical and the simulation:
The statistical analysis used is appropriate.
3. Results and Discussion:
The authors should be reduce this part, please the results shown in figures.
This part is very long, reduce it and discuss only the obtained results.
L236-485… … However, even abiotic conditions should not be overlooked.
Author Response
Comment 1:
It’s necessary to insert new references about the use of natural water see below reference.
Reply:
New references were added in the revised manuscript.
Comment 2:
L167-227… For the survival experiment, bacteria in the logarithmic growth phase were diluted and treated with 300 μM H2O2 or 550 μM diamide for 1 h, and then the colony-forming unites were counted. the data was collected (the experiment protocol was developed by you with due care? There was it mortality?
Reply:
After trying different concentrations of H2O2 and diamide, we found 300 μM H2O2 and 550 μM diamide were optimum concentration for our experiments. The mortlity is as follows:
Comment 3:
- Results and Discussion: The authors should be reduce this part, please the results shown in figures. This part is very long, reduce it and discuss only the obtained results. L236-485… … However, even abiotic conditions should not be overlooked.
Reply:
We have appropriately reduced the results and discussions. Lines 249-474.

Reviewer 2 Report
Comments to authors:
Line 17. Please, change “severe” by “important”.
· Lines 17-18, please, rewrite, the sentence is confusing
· Line 39, please, change “aquaculture, there..” by “aquaculture, therefore, there…”
· Line 41, please, change “as members” by “as a member”
· Line 53-54, please, rewrite, for example: It adapts well to different temperatures, salt concentration ...
· Lines 57,58, please, rewrite
· Table S1, please cite the species of fish from which the strain TX01 has been isolated.
· Line 201-202: Please briefly but concisely describe the mobility assay, for ease of reading, not just cite the reference.
· Lines 203-205: Please briefly but concisely describe the serum resistance assay, for ease of reading, not just cite the reference.
· Line 209. Please, rewrite: 37ºC. Different E. piscicida strains.
· Line 212. Please, provide the MIC concentration of this antibiotic for this particular strain of E. piscicida.
· Line 223-224. Please, enter the number of fish per group, number of groups, controls used, etc.
· Line 382-383. “At each time point, intracellular were determined. The viable bacteria were determined”. The authors mean that viable intracellular bacteria were listed. Please correct the sentence.
· Macrophage experiments. ¿Why the authors have chosen a mouse cell, which grows at 37ºC to perform experiments with a bacterium that is normally used at 26-28ºC?.
· Figure 5B. E. piscicida can replicate and survive within host macrophages, therefore, the experiment should be performed at longer times of infection in vitro, if not, the defect in the growth of mutants can be masked, since, according to Figure 3, there are important differences. The replication rate of E. piscicida in macrophages appears very slow. Please discuss this, because if there are fewer intracellular bacteria of some mutants at time 0h, it is logical that there are fewer bacteria at times of 2h and 4h.
· Figure 5B. The authors should provide some photographs of the infections, to know the exact location of the bacteria.
· Throughout the text, in vitro and in vivo should be italicized.
· Please describe and discuss the mortality observed in infected fish.
Author Response
Comment 1:
Correction of errors in word use, sentence meaning, and format.
Reply:
We have revised it in the manuscript. Lines 17-18, 40, 42, 54-55, 56-57, 387,.512.
Comment 2:
Table S1, please cite the species of fish from which the strain TX01 has been isolated.
Reply:
We redescribe the source of TX01 in “2.2. Bacteria, plasmids, and cells” and redescribe in Table S1. “E. piscicida TX01 was isolated from diseased Japanese flounder at fish farms in north China and was determined by 16S rRNA gene analysis [27].” Lines 94-96.
Comment 3:
Line 201-202: Please briefly but concisely describe the mobility assay, for ease of reading, not just cite the reference
Reply:
We have added the description in revised manuscript. Lines 205-208.
Comment 4:
Lines 203-205: Please briefly but concisely describe the serum resistance assay, for ease of reading, not just cite the reference.
Reply:
We have added the description in revised manuscript. Lines 211-217.
Comment 5:
Line 209. Please, rewrite: 37ºC. Different E. piscicida strains.
We have revised it in the manuscript. Line 221.
Comment 6:
Line 212. Please, provide the MIC concentration of this antibiotic for this particular strain of E. piscicida.
Reply:
We have revised it. the minimum inhibition concentration of gentamicin is 20 μg/mL. lines 224-225
Comment 7:
Line 223-224. Please, enter the number of fish per group, number of groups, controls used, etc.
Reply:
We have added the description in revised manuscript. “Tilapias were randomly divided into 5 groups and 10 per group, infected with 50 μL of each bacterial suspension or PBS (control) by intramuscular injection.” Lines 236-237.
Comment 8:
Line 382-383. “At each time point, intracellular were determined. The viable bacteria were determined”. The authors mean that viable intracellular bacteria were listed. Please correct the sentence.
Reply:
We have revised the sentence. “At each time point, the viable intracellular bacteria were determined.” Lines 383.
Comment 9:
Macrophage experiments. Why the authors have chosen a mouse cell, which grows at 37ºC to perform experiments with a bacterium that is normally used at 26-28ºC ?
Reply:
RAW264.7 are one of the host cells for E. piscicida and were widely used for the infection experiment of E. piscicida (Leung K Y et al., 2019). Since we have not obtained macrophage cells of fish, so RAW264.7 are used. According to our observation, there is no significant effect on RAW264.7 status in short-term 28°C incubation, and 28°C is advantageous for infection experiment of E. piscicida.
Leung K Y, Wang Q, Yang Z and Siame B A. Edwardsiella piscicida: a versatile emerging pathogen of fish. 2019, 10: 555-567.
Comment 10:
Figure 5B. E. piscicida can replicate and survive within host macrophages, therefore, the experiment should be performed at longer times of infection in vitro, if not, the defect in the growth of mutants can be masked, since, according to Figure 3, there are important differences. The replication rate of E. piscicida in macrophages appears very slow. Please discuss this, because if there are fewer intracellular bacteria of some mutants at time 0 h, it is logical that there are fewer bacteria at times of 2 h and 4 h.
Reply:
Most RAW264.7 cells can not completely adhere to the wall of plate after 6-8 h of incubation with the E. piscicida, so we did not delay the assay for the sake of experimental rigor.
It seems not reasonable to compare the two experiments of Figure 5 and Figure 3. These are two different culture environments, different experimental operations and the number of starting bacteria in these two experiments is different.
Comment 11:
Figure 5B. The authors should provide some photographs of the infections, to know the exact location of the bacteria.
Reply:
According to our previous observations (Xie J et al., 2021; Wang D et al., 2021), a large number of E. piscicida can survival within the cells. Considering too much figures in the manuscript, we did not take these corresponding laser confocal images
Xie, J.H.; Zhao, Q.; Huang, H.; Fang, Z.; Hu Y.H. Edwardsiella piscicida HigB: A type II toxin that is essential to oxidative re-sistance, biofilm formation, serum survival, intracellular propagation, and host infection. Aquaculture 2021, 535, 736382.
Wang D, Gong C, Gu H, Huang H, Xian J and Hu Y Bicistronic operon YhaO-YhaM contributes to antibiotic resistance and virulence of pathogen Edwardsiella piscicida. Aquaculture, 2021, 541: 736849.
Comment 12:
Please describe and discuss the mortality observed in infected fish.
Reply:
We have revised it in the manuscript. “Also, we did not observe fish mortality during the experiment because of the short duration. According to our laboratory data, infections with the same amount of WT generally show mortality after 4-5 days of injection [28, 37].” lines 392-393.

Reviewer 3 Report
The study adds basic knowledge necessary for pathogenicity of Edwardsiella piscicida, an important fish pathogen. The manuscript has a nice flaw and is methodologically very well organised. There are however a lot of mistakes in written English, a few of which are indicated below
Also a general comment that the authors should take into consideration is to provide some more info regrading the mutant types. Where are the mutations located? A figure might be very helpful.
Lines 38-40. The sentence is too naive. Several factors pose threats to aquaculture. Edwardsiellosis is one of them. Please rephrase
In line 45, please replace “seawater” with “marine”
Line 55: “Since its importance and characteristic of infection”, please rephrase, hard to understand and missing connectivity
Apart form the reference number 27 in line 93, the authors should add some details concerning the diseased fish
Line 446: Please replace “involve” with “involved”
Line 450: Please replace “exhibit” with “exhibits”
